# Different detection capabilities by mycological media for *Candida* isolates from mono- or dual-species cultures

Giulia De Angelis[1]☯, Giulia Menchinelli[1]☯, Riccardo Torelli[2], Elena De Carolis[2], Patrizia Posteraro[3], Maurizio Sanguinetti[1,2]*, Brunella Posteraro[4,5]

**1** Istituto di Microbiologia, Università Cattolica del Sacro Cuore, Rome, Italy, **2** Dipartimento di Scienze di Laboratorio e Infettivologiche, Fondazione Policlinico Universitario A. Gemelli IRCCS, Rome, Italy, **3** Laboratorio di Analisi Cliniche e Microbiologiche, GVM Ospedale San Carlo di Nancy, Rome, Italy, **4** Istituto di Patologia Medica e Semeiotica Medica, Università Cattolica del Sacro Cuore Rome, Rome, Italy, **5** Dipartimento di Scienze Gastroenterologiche, Endocrino-Metaboliche e Nefro-Urologiche, Fondazione Policlinico Universitario A. Gemelli IRCCS, Rome, Italy

☯ These authors contributed equally to this work.
* maurizio.sanguinetti@unicatt.it

**Data Availability Statement:** All relevant data are within the manuscript and its Supporting Information files.

## Abstract

The aim of this study was to compare the Candida bromcresol green (BCG) medium with the chromogenic (CHROM) Brilliance Candida agar and Sabouraud dextrose agar (SDA) media in regard to their capability of detecting *Candida* isolates from mono- or dual-species cultures. We prepared *Candida* isolates' suspensions to obtain mono-species (n = 18) or dual-species (n = 153) culture plates per each medium, and three readers independently observed 513 plates at 24-h, 48-h and 72-h incubation time. We scored reading results as correct, over or under detection compared to the expected species number(s). BCG showed significantly higher correct-detection and lower under-detection rates for all *Candida* species when observed by at least one reader. At 24-h reading, 12 mono-species cultures had correct (or over) detections in all media, whereas 106, 60 and 78 dual-species cultures had correct (or over) detections in BCG, CHROM or SDA, respectively. BCG provides the basis for an accurate laboratory diagnosis of *Candida* infections.

## Introduction

Almost concurrent with the enormous advances in medical diagnosis and treatment, a growing number of individuals have become susceptible to acquiring fungal infections [1], and the majority of these infections is lethal for more than 1.5 million people [2]. Fungal infections such as mucosal/skin infections, though non-lethal, can reduce the quality of life for >1 billion affected people [2]. As opportunistic fungi, *Candida* species are the prevalent causes of invasive (e.g. candidaemia) and non-invasive (e.g. vulvovaginal candidiasis) fungal diseases, with an estimated ~700,000 invasive candidiasis cases occurring annually [2].

Early diagnosis and, consequently, prompt treatment of invasive *Candida* infections is crucial to prevent mortality [3]. Five *Candida* species, *Candida albicans*, *Candida glabrata*,

**Funding:** This study was supported by Università Cattolica del Sacro Cuore (Fondi Ateneo, linea D1-2018).

**Competing interests:** The authors have declared that no competing interests exist.

*Candida tropicalis*, *Candida parapsilosis* and *Candida krusei*, are responsible for 92% of cases of candidaemia globally [4]. However, the gamut of clinically relevant *Candida* species is expanding [5] and, notably, less common species (i.e. *Candida guilliermondii* complex) [6], rare species (e.g. *Candida inconspicua* [*Torulopsis inconspicua*], *Candida pararugosa* and *Pichia norvegensis* [*Candida norvegensis*]) [7], or emerging species (i.e. *Candida auris*) [8] may exhibit high antifungal resistance levels, thereby compromising infection outcomes. Furthermore, mixed bloodstream infections with *Candida* species in single patient-episodes are not uncommon [6, 9], consequently leading to misdiagnoses because of apparently pure isolates in mycological cultures.

Implementation of matrix-assisted laser desorption ionization-time-of-flight (MALDI-TOF) mass spectrometry (MS) in clinical mycology diagnostics has greatly shortened the time for fungal species identification [10]. However, the accuracy of identification still relies on the precision of picking fungal colonies from primary culture plates (i.e. directly derived from clinical samples) and on the media used to enable fungal colony growth. Chromogenic media (e.g. Brilliance Candida agar; Oxoid, Thermo Scientific, Basingstoke, UK) are currently used for both isolating and/or presumptively identifying *Candida* species from primary cultures in clinical microbiology laboratories [11, 12]. By contrast, the traditional Sabouraud dextrose agar (SDA) medium (Vacutest Kima S.r.l.) allows to isolate from and differentiate *Candida* species in primary cultures based on macromorphology features. Since several years, our laboratory adopted the Candida bromcresol green (BCG) medium (Vacutest Kima S.r.l., Padua, Italy) [13] as an alternative to the SDA [9]. The BCG had been introduced from Difco Laboratories (Detroit, MI, USA) as a differential and selective medium for primary isolation and detection of *Candida* species from clinical samples. However, to the best of our knowledge, no study published did include the BCG medium in their mycological media evaluation.

We compared the performance of BCG medium with those of Brilliance Candida agar medium (hereafter referred to as CHROM medium) and SDA medium, using *Candida* species allowed to grow in pure (mono-species) or mixed (dual-species) cultures, respectively. In addition to the species claimed by the CHROM medium's manufacturer as presumptively identifiable (i.e. *C. albicans*, *C. krusei* and *C. tropicalis*) [12], we tested other species (including *C. auris*) to comprise five common and 13 uncommon species of *Candida* in total.

## Materials and methods

We used 18 selected *Candida* species that belonged to the clinical isolate collection hold at the Fondazione Policlinico Universitario A. Gemelli IRCCS, Rome (Italy). Only *C. auris* was obtained from the Center of Expertise in Mycology Radboudumc/CWZ, Nijmegen (The Netherlands). The species (isolate) included in the study were, in alphabetic order, *C. albicans* (UCSC34/23), *C. auris* (CWZ-1), *C. dubliniensis* (UCSC35/12), *C. glabrata* (UCSC61/2), *C. guilliermondii* (UCSC36/14), *C. incospicua* (UCSC72/2), *Candida kefyr* (UCSC51/14), *C. krusei* (UCSC59/12), *Candida lusitaniae* (UCSC59/18), *Candida nivariensis* (UCSC11/3), *C. norvegensis* (UCSC64/13), *C. parapsilosis* (UCSC30/27), *C. pararugosa* (UCSC35/20), *Candida pelliculosa* (UCSC72/2), *Candida robusta* (UCSC54/2), *Candida sorbosa* (UCSC28/45), *C. tropicalis* (UCSC49/29) and *Candida utilis* (UCSC36/21). All the isolates were from patient bloodstream infections. Before testing, we recovered isolates from frozen stocks by culture on SDA plates at 30˚C ensuring vitality and/or pure growth. To confirm their identity, isolates were re-identified using the MALDI-TOF MS based method, as previously described [14].

We used each isolate to prepare a 0.5 McFarland suspension (~$10^6$ CFU/ml) in phosphate-buffered saline [12]. Using a checkerboard-like dilution scheme (S1 Fig), we mixed (ratio 1:1) each isolate's suspension with the suspension of itself or with the suspension of each other

isolate, respectively, in Axygen® 96-Deep Well polypropylene plates to reach a 1-ml final volume for a concentration of ~500 CFU per well. To obtain mono-species (n = 18) or dual-species (n = 153) cultures per medium, we used a spatula to spread a 100-μl (~50 CFU) aliquot from each well on the surfaces of BCG, CHROM or SDA plates. We incubated 513 plates at 30˚C, according to the media manufacturers' instructions. Preliminarily, we performed controls with the isolates' suspensions to verify growth, number of CFU and identity of *Candida* species (single or multiple) expected to grow on the plates. Although the common *Candida* species grew well at 37˚C, at least on the BCG or SDA media, we chose the 30˚C incubation to favour the uncommon *Candida* species (the majority in the study) growing slowly at 37˚C (e.g. *C. guilliermondii*). Three authors independently read the plates after 24, 48 and 72 h of incubation, in a blinded manner regarding type(s) and number(s) of *Candida* species growing on plates. These conditions are those universally accepted for the isolation of medically important yeasts from clinical specimens [15], but simultaneous incubation at 30˚C and 37˚C may be useful [16].

We scored individual reader results daily regarding how many different colonies, in terms of morphologic appearance (including texture and/or colour), he/she was able to observe. For the BCG, CHROM or SDA plate series (i.e. three plates for each species or combination of species tested per medium), we recorded the number of colonies observed by the readers on each plate, and we compared the numbers obtained with those expected for each plate of the three series (S1 Table). Thus, we reported reading results as correct detections (when the number of observed species equalled the number of expected species), over detections (when the number of observed species exceeded the number of expected species) or under detections (when the number of observed species was inferior to the number of expected species). Although more colony morphotypes do not necessarily correspond to different species, we considered the term morphotype as the equivalent of species when recorded our reading results. If necessary, we stratified the reading results by all the species (n = 18), common species (n = 5) or uncommon species (n = 13) obtained with the *Candida* isolates grown in either mono-species or dual-species culture plates. We compared detection rates on the BCG versus CHROM or SDA plates using chi-square test. We considered a p value of <0.05 statistically significant. We used the weighted kappa coefficient to assess the inter-reader agreement, with ranges described in literature [17].

## Results

We obtained 54 mono-species and 459 dual-species *Candida* cultures (171 on BCG, 171 on CHROM and 171 on SDA plates), and the results of detecting *Candida* species are shown in Fig 1 (all species), S2 Fig (common species) and S3 Fig (uncommon species). The percentages of correct detections of the three *Candida* species groups, compared to expected results, ranged with the BCG medium from 73.1 (24 h) to 90.1 (72 h), 73.3 (48/72 h) to 93.3 (72 h), and 64.8 (24 h) to 91.2 (72 h); with the CHROM medium from 44.4 (24 h) to 90.6 (72 h), 53.3 (24 h) to 93.3 (48/72 h), and 40.7 (24 h) to 87.9 (72 h); and with the SDA medium from 60.8 (24 h) to 92.4 (72 h), 86.7 (24/48/72 h) to 93.3 (24/48/72 h), and 52.7 (24 h) to 91.2 (72 h). The percentages of over detections ranged with the BCG medium from 0.6 (24 h) to 9.4 (48 h), 6.7 (24/48/72 h) to 20.0 (24/48/72 h), and 0.0 (24 h) to 7.7 (72 h); with the CHROM medium from 0.6 (24/48/72 h) to 4.7 (48 h), 0.0 (48/72 h) to 13.3 (48/72 h), and 0.0 (24 h) to 4.4 (24/72 h); and with the SDA medium from 0.0 (24 h) to 4.7 (48/72 h), 0.0 (24 h) to 13.3 (48/72 h), and 0.0 (24/48/72 h) to 6.6 (72 h). The percentages of under detections ranged with the BCG medium from 2.9 (72 h) to 26.3 (24 h), 0.0 (24/48/72 h) to 6.7 (24/48/72 h), and 1.1 (72 h) to 35.2 (24 h); with the CHROM medium from 5.8 (72 h) to 55.0 (24 h), 6.7 (48/72 h) to 40.0 (24 h), and 7.7

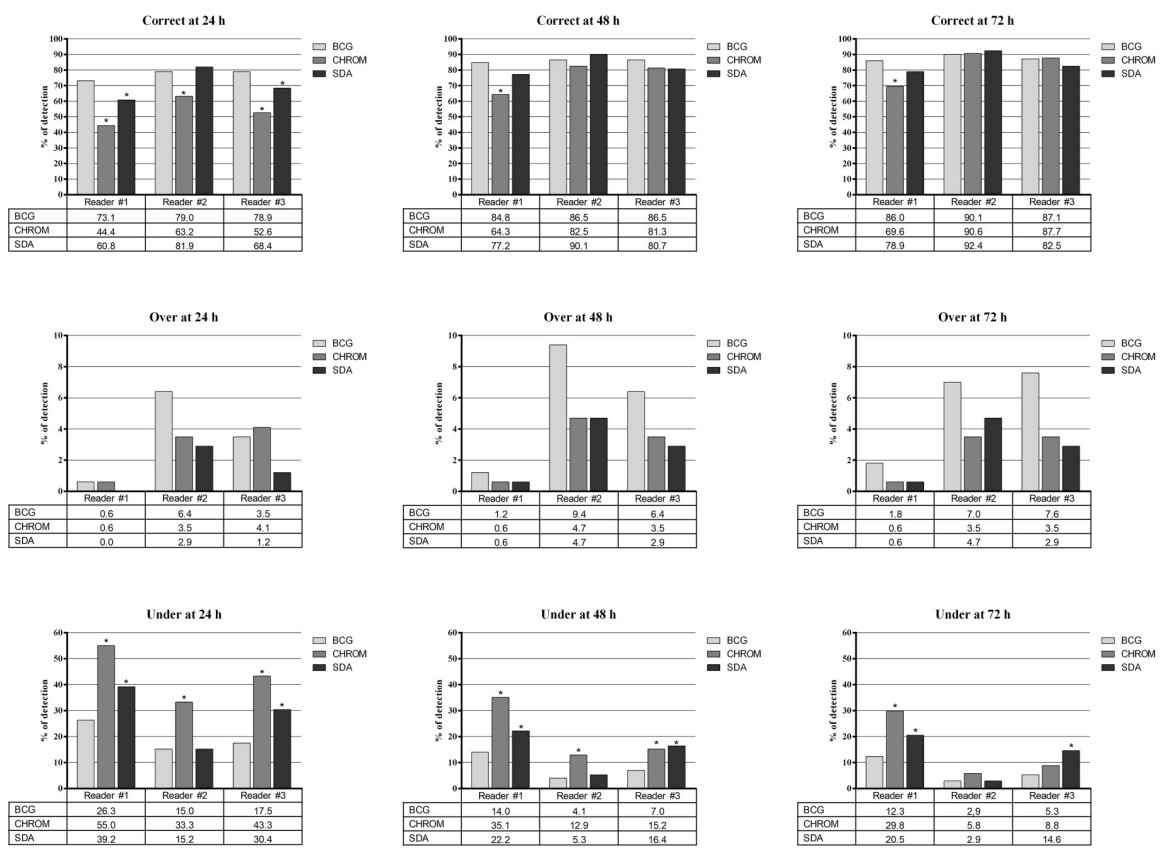

**Fig 1. Rates of correct, over or under detections by three readers for the overall *Candida* species grown as mono- (n = 18) or dual-species cultures (n = 153) on the BCG (Candida bromcresol green), CHROM (chromogenic medium, i.e. Brilliance Candida agar) and SDA (Sabouraud dextrose agar) media.** Asterisks indicate statistically significant differences between the rates of detections obtained with the BCG medium and those of the CHROM or SDA media.

(72 h) to 59.3 (24 h); with the SDA medium from 2.9 (72 h) to 39.2 (24 h), 0.0 (24/48/72 h) to 6.7 (24 h), and 2.2 (72 h) to 47.3 (24 h).

As for all the *Candida* species observed by at least one reader, statistically significant differences in the rates of correct or under detections did favour the BCG medium over the CHROM medium (24/48/72 h) and the SDA medium (24 h and 24/48/72 h, respectively) (Fig 1). As for the common *Candida* species observed by at least one reader, statistically significant differences in the rates of correct or under detections did favour the BCG medium over the CHROM medium (24 h) (S2 Fig). As for the uncommon *Candida* species observed by at least one reader, statistically significant differences did favour the BCG medium over the CHROM medium in the rates of correct detections (24/48/72 h) and over both the CHROM and SDA media in the rates of under detections (24/48/72 h) (S3 Fig).

As shown in Table 1, we analysed detection results regarding inter-reader agreement. The levels of agreement for the 24-h readings of BCG, CHROM and SDA plates, for all or uncommon species, were at least moderate (kappa coefficient values, 0.41–0.60), whereas the levels of agreement for the 48-h and 72-h readings of BCG, CHROM and SDA plates were at least fair (kappa coefficient values, 0.21–0.40). Conversely, for common species, the levels of agreement for the 24-h/48-h/72-h readings of CHROM plates and for the 48-h/72-h readings of SDA plates were at least substantial (kappa coefficient values, 0.61–0.80). However, comparing the three readers with respect to the percentages of incorrect (over/under) detections for overall

**Table 1. Agreement by readers on the detection results of *Candida* species cultured on three media plates that were obtained at 24, 48 or 72 h of incubation of the plates.**

| *Candida* organisms grown in mono- or dual-species cultures | Culture media plates read at the indicated times | | N. of overall detections | N. of detections found to be in agreement and scored as | | | Kappa coefficient (95% CI) | Level of inter-reader agreement |
|---|---|---|---|---|---|---|---|---|
| | | | | Correct | Over | Under | | |
| All species (n = 18) | BCG | 24 h | 171 | 108 | 1 | 18 | 0.53 (0.41–0.64) | Moderate |
| | CHROM | | 171 | 59 | 0 | 52 | 0.55 (0.47–0.64) | Moderate |
| | SDA | | 171 | 87 | 0 | 26 | 0.47 (0.37–0.58) | Moderate |
| | BCG | 48 h | 171 | 120 | 1 | 4 | 0.29 (0.15–0.43) | Fair |
| | CHROM | | 171 | 99 | 1 | 19 | 0.47 (0.36–0.59) | Moderate |
| | SDA | | 171 | 113 | 1 | 7 | 0.34 (0.22–0.48) | Fair |
| | BCG | 72 h | 171 | 126 | 2 | 0 | 0.25 (0.13–0.38) | Fair |
| | CHROM | | 171 | 109 | 1 | 7 | 0.30 (0.18–0.44) | Fair |
| | SDA | | 171 | 119 | 1 | 3 | 0.31 (0.19–0.43) | Fair |
| Common species only (n = 5) | BCG | 24 h | 15 | 12 | 1 | 0 | 0.59 (-0.04–1.30) | Moderate |
| | CHROM | | 15 | 7 | 0 | 5 | 0.75 (0.48–1.04) | Substantial |
| | SDA | | 15 | 12 | 0 | 0 | 0.20 (-0.29–0.86) | Fair |
| | BCG | 48 h | 15 | 9 | 1 | 0 | 0.28 (-0.43–1.16) | Fair |
| | CHROM | | 15 | 12 | 0 | 1 | 0.68 (0.18–1.27) | Substantial |
| | SDA | | 15 | 13 | 1 | 0 | 0.72 (-0.41–2.76) | Substantial |
| | BCG | 72 h | 15 | 10 | 1 | 0 | 0.36 (-0.41–1.37) | Fair |
| | CHROM | | 15 | 12 | 0 | 1 | 0.63 (-0.03–1.52) | Substantial |
| | SDA | | 15 | 13 | 1 | 0 | 0.72 (-0.41–2.76) | Substantial |
| Uncommon species only (n = 13) | BCG | 24 h | 91 | 53 | 0 | 15 | 0.59 (0.45–0.74) | Moderate |
| | CHROM | | 91 | 26 | 0 | 28 | 0.49 (0.37–0.61) | Moderate |
| | SDA | | 91 | 40 | 0 | 20 | 0.52 (0.39–0.66) | Moderate |
| | BCG | 48 h | 91 | 67 | 0 | 4 | 0.39 (0.18–0.62) | Fair |
| | CHROM | | 91 | 47 | 1 | 15 | 0.54 (0.40–0.68) | Substantial |
| | SDA | | 91 | 56 | 0 | 4 | 0.28 (0.11–0.46) | Fair |
| | BCG | 72 h | 91 | 71 | 1 | 0 | 0.33 (0.17–0.51) | Fair |
| | CHROM | | 91 | 54 | 1 | 5 | 0.35 (0.19–0.53) | Fair |
| | SDA | | 91 | 57 | 0 | 2 | 0.23 (0.08–0.38) | Fair |

Mycological media used for mono- or dual-species cultures of all, common or uncommon species of *Candida* were BCG (Candida bromcresol green), CHROM (chromogenic medium, i.e. Brilliance Candida agar) and SDA (Sabouraud dextrose agar). We calculated the kappa coefficient with 95% confidence interval (95% CI) for the comparison among the rates of correct, over or under detections (according to the definitions specified in the text). With regard to the agreement by readers, we used values greater than zero to indicate none to slight (0.01–0.20), fair (0.21–0.40), moderate (0.41–0.60), substantial (0.61–0.80) or almost perfect (0.81–1.00) levels of agreement, and values lower than/equal to zero to indicate the absence of agreement

readings of BCG, CHROM and SDA plates revealed statistically significant differences between the readers only for CHROM and SDA plates (S2 Table).

The BCG medium is similar to other selective and differential media for the primary isolation of *Candida* species. It consists of peptone agar base supplemented with yeast extract (which is absent in the SDA) and dextrose to support growth [13]. However, bromcresol green helps to differentiate and identify *Candida* species, because a change in the pH causes the medium to take on a yellow colour around the *Candida* colonies that ferment dextrose [13]. Fig 2 depicts the appearance on the BCG medium for the five common species of *Candida* tested by us. As it can see, *C. albicans* formed smooth, regular, matte, and white to dark-green colonies, *C. glabrata* smooth, circular, brilliant and white to pale-green colonies,

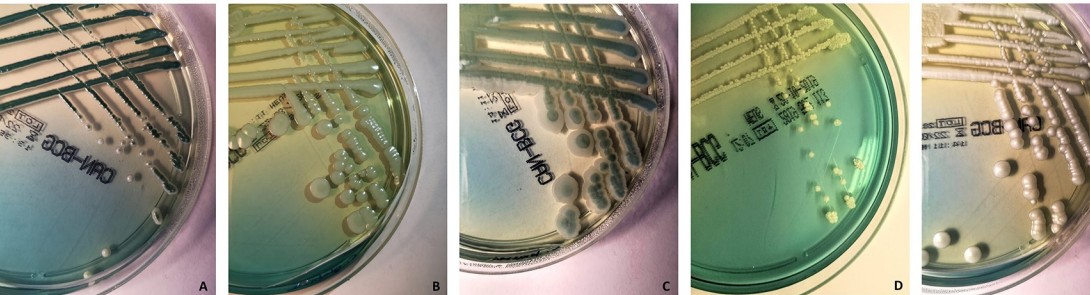

**Fig 2. Appearance on the Candida bromcresol green (BCG) medium of five *Candida* isolates included in the study that belong to (A) *C. albicans*, (B) *C. glabrata*, (C) *C. krusei*, (D) *C. parapsilosis* and (E) *C. tropicalis*.** The isolates were seeded on the BCG plates and incubated at 30˚C before the plates were imaged. (See the text for the detailed description of the isolates' features).

*C. krusei* rough, irregular, matte and green and white-edged colonies, *C. parapsilosis* rough, irregular, small and white colonies, and *C. tropicalis* smooth, regular, matte and white colonies.

## Discussion

Apart from the overall slight superiority shown by the BCG medium, we noticed that 12 (66.7%) of 18 mono-species cultures at the 24-h readings had correct or over detections (i.e. cultures with ≥1 colony morphologies observed per plate) in the BCG plates as well in both the CHROM and SDA plates (S1 Table). The only exceptions were *C. albicans*, *C. incospicua*, *C. lusitaniae*, *C. pararugosa*, *C. pelliculosa* and *C. sorbosa*. Interestingly, while detection of *C. albicans* in the CHROM plates occurred not prior to 48 h of incubation, over detection of *C. glabrata* occurred always in 100% of BCG plate readings, 55.6% of CHROM plate readings, and 88.9% of SDA plate readings. These results are consistent with those from some previous studies [12, 18]. In one study, 132 (23%) of the 564 *C. albicans* isolates recovered by routinely used media, did not grow on a chromogenic medium [18]. Another study showed the presumptive identification of five *Candida* species (*C. albicans*, *C. dubliniensis*, *C. krusei*, *C. tropicalis* and *C. parapsilosis*) on the CHROM medium (i.e. two additional species besides those identifiable by the medium) [12].

However, discrimination for several *Candida* species, including *C. glabrata*, may be difficult. Conversely, the bromcresol green, a non-toxic indicator contained in the BCG medium (i.e. a modified SDA), seems to aid primary isolation and detection of *Candida* species from clinical samples based on dextrose fermentation [13]. While the medium colour around the colonies becomes yellow (usually within 72 h of incubation), the *Candida* species grown on the BCG medium produce convex to cone-shaped, smooth to rough colonies. Thus, the BCG medium would allow to easily revealing differences in colour (i.e. tonalities of yellow) as well in morphology (i.e. extents of roughness) (Fig 2). We noted that one reader differed from the two other readers with respect to the over detection at 24, 48 and 72 h mainly for the uncommon *Candida* species (S3 Fig). Although we chose the three readers to represent a medium-to-high extent of expertise in medical mycology, it is plausible that subtle differences in their mycological skills may explain for the moderate or fair levels of agreement found across readers (Table 1).

Among mixed candidaemia episodes, *C. albicans* plus *C. glabrata* is usually the most frequent combination [5], but other combinations of *Candida* species may be of great importance. We noticed that 106 (69.3%), 60 (39.2%) and 78 (51.0%) of 153 dual-species cultures at

the 24-h readings had correct or over detections (i.e. cultures with ≥2 colony morphologies observed per plate) in the BCG, CHROM or SDA plates, respectively (S1 Table). More interestingly, six cultures (*C. albicans*/*C. parapsilosis*, *C. incospicua*/*C. pelliculosa*, *C. incospicua*/*C. sorbosa*, *C. nivariensis*/*C. pelliculosa*, *C. pararugosa*/*C. sorbosa* and *C. sorbosa*/*C. tropicalis*) in the CHROM medium and two cultures (*C. auris*/*C. guilliermondii* and *C. auris*/*C. lusitaniae*) in the SDA medium were always under detected with respect to one of the two species grown together. Although a mix of these species seems to be very uncommon, their incomplete detection in candidaemia cases may have clinical repercussions, especially because of different antifungal susceptibility profiles exhibited by these species [5–8].

In conclusion, reading primary culture plates from patient samples in the clinical mycology laboratory remains somewhat subjective. For *Candida* species, the existence of different morphotypes, which underpins transitions between commensal and pathogenic cell types in the same species [19–21], complicates the situation. However, distinguishing as many as possible *Candida* colonies, which will likely correspond to different *Candida* species, in clinical samples, is crucial in order to exploit the established, powerful MALDI-TOF MS capability of identifying any *Candida* organism to the species level (or even beyond). Therefore, using the BCG medium may represent an essential prerequisite for a specific and accurate diagnosis of the causative infection agent(s), especially in patients suffering from life-threatening candidiasis.

## Supporting information

**S1 Table. Results of the detections by three independent readers for mono-species or dual-species *Candida* cultures recorded at 24, 48 or 72 h of incubation of the BCG, CHROM or SDA culture plates.**
(DOC)

**S2 Table. Rates of incorrect (over/under) detections by three readers for the overall BCG, CHROM or SDA cultures of *Candida* species.** Bold indicates statistically significant differences in the comparisons between reader #1 and reader #2, reader #2 and reader #3, or reader #1 and reader #3.
(DOC)

**S1 Fig. Scheme of the checkerboard-like dilution method used to obtain mono-species or dual-species *Candida* suspensions before spreading them on the mycological media under evaluation.**
(DOC)

**S2 Fig. Rates of correct, over or under detections by three readers for the common *Candida* species grown as mono- (n = 5) or dual-species (n = 10) on the BCG (Candida bromcresol green), CHROM (chromogenic medium, i.e. Brilliance Candida agar) and SDA (Sabouraud dextrose agar) media.** Asterisks indicate statistically significant differences between the rates of detections obtained with the BCG medium and those of the CHROM or SDA media.
(JPG)

**S3 Fig. Rates of correct, over or under detections by three readers for the uncommon *Candida* species grown as mono- (n = 13) or dual-species (n = 78) on the BCG (Candida bromcresol green), CHROM (chromogenic medium, i.e. Brilliance Candida agar) and SDA (Sabouraud dextrose agar) media.** Asterisks indicate statistically significant differences between the rates of detections obtained with the BCG medium and those of the CHROM or SDA media.
(JPG)

## Acknowledgments

We wish to thank Franziska Lohmeyer for her English language assistance.

## Author Contributions

**Conceptualization:** Giulia De Angelis, Brunella Posteraro.

**Data curation:** Giulia De Angelis, Giulia Menchinelli, Riccardo Torelli, Elena De Carolis, Patrizia Posteraro, Brunella Posteraro.

**Formal analysis:** Giulia Menchinelli, Riccardo Torelli, Elena De Carolis, Patrizia Posteraro.

**Investigation:** Giulia Menchinelli, Riccardo Torelli, Elena De Carolis, Patrizia Posteraro.

**Methodology:** Giulia Menchinelli, Brunella Posteraro.

**Supervision:** Maurizio Sanguinetti.

**Validation:** Maurizio Sanguinetti.

**Writing – original draft:** Giulia De Angelis, Brunella Posteraro.

**Writing – review & editing:** Brunella Posteraro.

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
