## [Decision Letter · Decision Letter 0]

14 Jan 2020

PONE-D-19-32923

Different detection capabilities by mycological media for Candida isolates from mono- or dual-species cultures

PLOS ONE

Dear Dr. Sanguinetti,

Thank you for submitting your manuscript to PLOS ONE. After careful consideration, we feel that it has merit but does not fully meet PLOS ONE’s publication criteria as it currently stands. Therefore, we invite you to submit a revised version of the manuscript that addresses the points raised during the review process.

We would appreciate receiving your revised manuscript by Feb 28 2020 11:59PM. To enhance the reproducibility of your results, we recommend that if applicable you deposit your laboratory protocols in protocols.io, where a protocol can be assigned its own identifier (DOI) such that it can be cited independently in the future. For instructions see: http://journals.plos.org/plosone/s/submission-guidelines#loc-laboratory-protocols

We look forward to receiving your revised manuscript.

Kind regards,

Alex Friedrich

Academic Editor

PLOS ONE

Journal Requirements:

2. For reproducibility purposes, please provide the ID # for all clinical isolates used in this analysis.

'This study was supported by Università Cattolica del Sacro Cuore (Fondi Ateneo, linea D1-

2018).'

'The author(s) received no specific funding for this work.'

Please provide an amended Funding Statement that declares *all* the funding or sources of support received during this specific study (whether external or internal to your organization) as detailed online in our guide for authors at http://journals.plos.org/plosone/s/submit-nowPlease state what role the funders took in the study.  If any authors received a salary from any of your funders, please state which authors and which funder. If the funders had no role, please state: "The funders had no role in study design, data collection and analysis, decision to publish, or preparation of the manuscript."

Reviewers' comments:

Reviewer's Responses to Questions

**Comments to the Author**

1. Is the manuscript technically sound, and do the data support the conclusions?

Reviewer #1: Yes

Reviewer #2: Yes

2. Has the statistical analysis been performed appropriately and rigorously? 

Reviewer #1: Yes

Reviewer #2: Yes

3. Have the authors made all data underlying the findings in their manuscript fully available?

Reviewer #1: Yes

Reviewer #2: Yes

4. Is the manuscript presented in an intelligible fashion and written in standard English?

Reviewer #1: Yes

Reviewer #2: Yes

5. Review Comments to the Author

Reviewer #1: In this work the Authors compared the performance of 3 different mycological media, namely BCG, CHROM and the non chromogenic SDA, for the detection of different Candida species cultures, both in mono-species and dual-species culture plates. Plates were read at 24, 48 and 72 hrs from 3 indipendent readers, and results were interpreted as correct detection, over detection, or under detection. Overall, BCG medium showed the best performance with higher rates of correct detection and lower rates of under detection for all Candida species analyzed, but also by splitting Candida spp in common and uncommon species, although with differences in the statistical significance. The Authors conclude BCG medium may represent an accurate tool for the correct identification to the species level and for the laboratory diagnosis of candidemia.

The matter of the manuscript is of interest in clinical diagnostics and provides valuable indications for the microbiology laboratory in the accurate diagnosis of Candida infections.

Some considerations:

- From what stated in the text, it would be interesting to know the opinion for the right time and temperature of incubation of plates for clinical specimens for the isolation of Candida spp, also as indicated by the literature.

- It is not clear if the term “over detection” means culture with more colony morphotypes, but not necessarily different species, or more Candida species than those expected. Please, clarify it.

- It would be interesting to know the true clinical impact of mixed Candida infections, specifically in candidemia (page 8, lines 170-174, the mixed cultures reported seems to be very uncommon, right?)?

- page 5, line 84-88. Please, it is necessary to better specify the final volume (the final concentration) reached in each well. Could the Authors provide some references to support their choice for the methodologies?

- Page 8, lines 161-165. I would add this part also in the Results section along with the reference of Fig 4, with a brief description of the characteristics of the Candida media used.

- The resolution of Figures 1, 2, 3, needs to be much improved, they are blurred. Moreover, it is also difficult to notice what the Authors state for the Figure 4 with regard to the differences in the colony morphologies.

- There is a variability between reader 1 and the others respect to the over detection at 24, 48 and 72 hrs mainly for all Candida species and the uncommon Candida species, as indicated in the Figures 1 to 3 and in Table 1 (moderate or fair levels of agreement).What is the reason of it? Does this could affect the results?

Reviewer #2: The ms by De Angelis et al is a comparison study among different commercially available media (three) in order to understand if these media can be used for detecting different Candida spp from primary cultures.

The AA used 18 strains belonging to different species of Candida, all isolates from BSI. A C.auris strain was also included. Following a CK scheme, cultures were variably combined to obtain mono, dual and triple combination of strains. All these unique combinations were plated on the three media and results were reported in 1 table and many figures.

The ms is clearly written and results obtained , even with limitations described by the AA (reading plates remain very subjective, and the different morphotype of Candida) are in favor of the use of BCG.

In my opinion, two minor points need to be addressed:

1) the measure of the level of inter reader agreement is somewhat vague. Any comment from the AA? Is it possible to measure this agreement in term of errors?

2) Number of figures can be reduced, summarizing the most significative results without losing the meaning of the message (the original figures can be added at the supplemental material section)

6. PLOS authors have the option to publish the peer review history of their article (what does this mean?). If published, this will include your full peer review and any attached files.

Reviewer #1: Yes: Giovanni Gherardi

Reviewer #2: No

---

## [Author Response · Author response to Decision Letter 0]

27 Jan 2020

PONE-D-19-32923

1. Please ensure that your manuscript meets PLOS ONE's style requirements, including those for file naming. The PLOS ONE style templates can be found at http://www.journals.plos.org/plosone/s/file?id=wjVg/PLOSOne_formatting_sample_main_body.pdf
http://www.journals.plos.org/plosone/s/file?id=ba62/PLOSOne_formatting_sample_title_authors_affiliations.pdf

Answer: We ensured that our manuscript was in line with the PLOS ONE's style requirements. See the revised manuscript throughout.

2. For reproducibility purposes, please provide the ID # for all clinical isolates used in this analysis.

Answer: We provided the ID # for all the clinical isolates used in our analysis. See the text and the footnote of S1 Fig.

'This study was supported by Università Cattolica del Sacro Cuore (Fondi Ateneo, linea D1-2018).'

'The author(s) received no specific funding for this work.'

Answer: We apologize for the incongruence of our statements. You will find the updated statements in the Funding Section.

Reviewer #1: In this work, the Authors compared the performance of 3 different mycological media, namely BCG, CHROM and the non-chromogenic SDA, for the detection of different Candida species cultures, both in mono-species and dual-species culture plates. Plates were read at 24, 48 and 72 hrs from 3 independent readers, and results were interpreted as correct detection, over detection, or under detection. Overall, BCG medium showed the best performance with higher rates of correct detection and lower rates of under detection for all Candida species analyzed, but also by splitting Candida spp. in common and uncommon species, although with differences in the statistical significance. The Authors conclude BCG medium may represent an accurate tool for the correct identification to the species level and for the laboratory diagnosis of candidemia.

The matter of the manuscript is of interest in clinical diagnostics and provides valuable indications for the microbiology laboratory in the accurate diagnosis of Candida infections.

Some considerations:

- From what stated in the text, it would be interesting to know the opinion for the right time and temperature of incubation of plates for clinical specimens for the isolation of Candida spp, also as indicated by the literature.

Answer: We provided details about the optimal time and temperature of incubation of plates for clinical specimens that are required for the isolation of Candida species. See lines 100–102 of the revised manuscript.

- It is not clear if the term “over detection” means culture with more colony morphotypes, but not necessarily different species, or more Candida species than those expected. Please, clarify it.

Answer: We clarified this issue adding a sentence as appropriate. See lines 111–113 of the revised manuscript.

- It would be interesting to know the true clinical impact of mixed Candida infections, specifically in candidemia (page 8, lines 170-174, the mixed cultures reported seems to be very uncommon, right?).

Answer: We added a comment about this interesting aspect raised by the reviewer. See lines 199–201 of the revised manuscript.

- page 5, line 84-88. Please, it is necessary to better specify the final volume (the final concentration) reached in each well. Could the Authors provide some references to support their choice for the methodologies?

Answer: We specified the final volume reached in each well as well as provided a reference to support our methodological choice. See lines 88–92 of the revised manuscript.

- Page 8, lines 161-165. I would add this part also in the Results section along with the reference of Fig 4, with a brief description of the characteristics of the Candida media used.

Answer: We added the part indicated by the reviewer in the Results section along with the reference of Fig 2 (formerly Fig 4) and a brief description of the media’s characteristics. See lines 155–164 of the revised manuscript.

- The resolution of Figures 1, 2, 3 needs to be much improved, they are blurred. Moreover, it is also difficult to notice what the Authors state for the Figure 4 with regard to the differences in the colony morphologies.

Answer: We improved all the Figures, namely Fig 1, S1 Fig (formerly Fig 2) and S2 Fig (formerly Fig 3) for quality. In particular, we redone the image of Fig 2 (formerly Fig 4) to make more evident the differences in the colony morphologies of the isolates photographed.

- There is a variability between reader 1 and the others respect to the over detection at 24, 48 and 72 hrs mainly for all Candida species and the uncommon Candida species, as indicated in the Figures 1 to 3 and in Table 1 (moderate or fair levels of agreement).What is the reason of it? Does this could affect the results?

Answer: We added a comment about this interesting aspect raised by the reviewer. See lines 185–189 of the revised manuscript.

Reviewer #2: The ms by De Angelis et al is a comparison study among different commercially available media (three) in order to understand if these media can be used for detecting different Candida spp from primary cultures.

The AA used 18 strains belonging to different species of Candida, all isolates from BSI. A C. auris strain was also included. Following a CK scheme, cultures were variably combined to obtain mono, dual and triple combination of strains. All these unique combinations were plated on the three media and results were reported in 1 table and many figures.

The ms is clearly written and results obtained, even with limitations described by the AA (reading plates remain very subjective, and the different morphotype of Candida) are in favor of the use of BCG.

In my opinion, two minor points need to be addressed:

1) The measure of the level of inter reader agreement is somewhat vague. Any comment from the AA?

Is it possible to measure this agreement in term of errors?

Answer: We agree with the reviewer on the vagueness of measuring the level of inter-reader agreement. Unfortunately, we were unable to measure the agreement in terms of errors. However, we added data about the rates of incorrect (over/under) detections of the three readers in order to underline the statistically significant differences between the readers with respect to the incorrect reading results. See lines 151–154 and a new Table in the Supporting information (S2 Table) of the revised manuscript.

2) Number of figures can be reduced, summarizing the most significative results without losing the meaning of the message (the original figures can be added at the supplemental material section).

Answer: As suggested, we move Fig 2 (now S1 Fig) and Fig 3 (now S2 Fig) to Supporting information.

---

## [Decision Letter · Decision Letter 1]

5 Mar 2020

Different detection capabilities by mycological media for Candida isolates from mono- or dual-species cultures

PONE-D-19-32923R1

Dear Dr. Sanguinetti,

We are pleased to inform you that your manuscript has been judged scientifically suitable for publication and will be formally accepted for publication once it complies with all outstanding technical requirements.

With kind regards,

Alex Friedrich

Academic Editor

PLOS ONE

Additional Editor Comments (optional):

Reviewers' comments:

Reviewer's Responses to Questions

**Comments to the Author**

1. If the authors have adequately addressed your comments raised in a previous round of review and you feel that this manuscript is now acceptable for publication, you may indicate that here to bypass the “Comments to the Author” section, enter your conflict of interest statement in the “Confidential to Editor” section, and submit your "Accept" recommendation.

Reviewer #1: All comments have been addressed

Reviewer #2: All comments have been addressed

2. Is the manuscript technically sound, and do the data support the conclusions?

Reviewer #1: Yes

Reviewer #2: Yes

3. Has the statistical analysis been performed appropriately and rigorously? 

Reviewer #1: Yes

Reviewer #2: Yes

4. Have the authors made all data underlying the findings in their manuscript fully available?

Reviewer #1: Yes

Reviewer #2: No

5. Is the manuscript presented in an intelligible fashion and written in standard English?

Reviewer #1: Yes

Reviewer #2: Yes

6. Review Comments to the Author

Reviewer #1: All comments have been adequately addressed. The revision version of the manuscript is much improved. This is an interesting manuscript and the entire manuscript is clear, concise and well written, the results obtained and discussion are strongly supported by the experimental data, with the overall high quality and appropriateness of the testing performed.

Reviewer #2: The AA addressed all my comments and in my opinion the manuscript is now acceptable for publication.

7. PLOS authors have the option to publish the peer review history of their article (what does this mean?). If published, this will include your full peer review and any attached files.

Reviewer #1: Yes: Giovanni Gherardi

Reviewer #2: No

---

## [Editor Report · Acceptance letter]

9 Mar 2020

PONE-D-19-32923R1 

Different detection capabilities by mycological media for *Candida* isolates from mono- or dual-species cultures 

Dear Dr. Sanguinetti:

I am pleased to inform you that your manuscript has been deemed suitable for publication in PLOS ONE. Congratulations! Your manuscript is now with our production department. 

With kind regards,

on behalf of

Dr. Alex Friedrich 

Academic Editor

PLOS ONE